# Three Efficient All-Erasure Decoding Methods for Blaum–Roth Codes

**DOI:** 10.3390/e24101499

**Published:** 2022-10-20

**Authors:** Weijie Zhou, Hanxu Hou

**Affiliations:** 1School of Computer Science and Technology, Dongguan University of Technology, Dongguan 523820, China; 2School of Electrical Engineering and Intelligentization, Dongguan University of Technology, Dongguan 523820, China

**Keywords:** distributed storage, Blaum–Roth codes, all-erasure decoding, decoding complexity

## Abstract

Blaum–Roth Codes are binary maximum distance separable (MDS) array codes over the binary quotient ring $F_{2}\left[x\right]/\left(M_{p}\left(x\right)\right)$, where $M_{p}\left(x\right)=1+x+\cdots +x^{p-1}$, and *p* is a prime number. Two existing all-erasure decoding methods for Blaum–Roth codes are the syndrome-based decoding method and the interpolation-based decoding method. In this paper, we propose a modified syndrome-based decoding method and a modified interpolation-based decoding method that have lower decoding complexity than the syndrome-based decoding method and the interpolation-based decoding method, respectively. Moreover, we present a fast decoding method for Blaum–Roth codes based on the LU decomposition of the Vandermonde matrix that has a lower decoding complexity than the two modified decoding methods for most of the parameters.

## 1. Introduction

Redundancy is necessary in storage systems in order to provide high data reliability in case of disk failures [1]. Replication and erasure codes are two main ways of including redundancy. The idea of replication is that the data in one disk are copied to multiple disks. The storage system replaces damaged disks with their copies when some disks are erased. It is fast to repair the erased disks but requires a lot of storage space. In contrast, erasure codes provide higher data reliability with a small storage cost.

Maximum distance separable (MDS) codes [2] are typical erasure codes that have optimal tradeoff between storage cost and data reliability, i.e., they can achieve the minimum storage cost given a level of data reliability. Binary MDS codes are special MDS codes that have lower computational complexity in the encoding/decoding procedures, since only XORs and cyclic-shift operations are involved. Some existing constructions of binary MDS codes are EVENODD codes [3,4], RDP codes [5], and X-codes [6,7], which can correct any two-column (we use “column" and “disk" interchangeably in this paper) erasures. RTP codes [8], Star codes [9,10], and extended EVENODD codes [11,12,13,14] can correct any three-column erasures. With the rapid increase in the data scale in storage systems [15], we need to design binary MDS codes that can correct any number of erasures as well as efficient encoding/decoding methods. Graftage codes [16] can achieve various tradeoffs between storage and repair bandwidth, while we focus on efficient decoding methods of binary MDS codes. Blaum–Roth codes [17] are this type of code, which are designed over the ring $R_{p}=F_{2}\left[x\right]/\left(M_{p}\left(x\right)\right)$, where $M_{p}\left(x\right)=1+x+\cdots +x^{p-1}$, and *p* is a prime number.

When some columns are erased, the syndrome-based decoding method [17] and the interpolation-based decoding method [18] have been proposed to recover the erased columns. In the decoding methods [17,18], there are three basic operations over the ring $R_{p}$: (i) addition, (ii) multiplication of a power of *x* and a polynomial, and  (iii) division of factor $1+x^{b}$ with 1≤b≤p−1. It is shown in the decoding methods [17,18] that we can first take the operations (i) and (ii) modulo $1+x^{p}$ and then take the results of modulo $M_{p}\left(x\right)$, while operation (iii) in the decoding methods [17,18] is directly taken as modulo $M_{p}\left(x\right)$.

In this paper, we show that we can also compute operation (iii) as modulo $1+x^{p}$, which has lower computational complexity than modulo $M_{p}\left(x\right)$. We propose modified decoding methods for the two existing decoding methods [17,18] that have a lower decoding complexity than the original decoding methods by computing operation (iii) as modulo $1+x^{p}$ instead of modulo $M_{p}\left(x\right)$. The reason our modified decoding methods have much lower decoding complexity than the decoding methods [17,18] is twofold. First, all the operations in our decoding methods are taken as modulo $1+x^{p}$, while the existing decoding methods execute the divisions as modulo $M_{p}\left(x\right)$. Second, we propose new algorithms in the decoding procedure to reduce the number of operations. Please refer to Section 3 for our two modified decoding methods. Moreover, the efficient LU decoding method [19] proposed for extended EVENODD codes decoding can also be employed to recover the erased columns of Blaum–Roth codes. We show that the LU decoding method has lower decoding complexity than the two modified decoding methods for most of the parameters. We define the decoding complexity as the total number of XORs required to recover the erased columns.

## 2. Blaum–Roth Codes

In this section, we first review the construction of Blaum–Roth codes [17] and then show the efficient operations over the ring $F_{2}\left[x\right]/\left(1+x^{p}\right)$. Finally, we present an algorithm to compute multiple multiplications, which have two nonzero terms over $F_{2}\left[x\right]/\left(1+x^{p}\right)$ with lower complexity.

### 2.1. Construction of Blaum–Roth Codes [17]

The codeword of Blaum–Roth codes [17] is a (p−1)×n array ${\left[c_{i,j}\right]}_{i=0,j=0}^{p-2,n-1}$ that is encoded from the (p−1)k information bits, where $c_{i,j}\in F_{2}$ and n≤p. We can view any *k* columns of the (p−1)×n array as information columns that store the (p−1)k information bits and the other r=n−k columns as parity columns that store the (p−1)r parity bits. For j=0,1,…,n−1, we represent the p−1 bits in column *j* by a polynomial $c_{j}\left(x\right)=\sum_{i=0}^{p-2}c_{i,j}x^{i}$. The (p−1)×n array of Blaum-Roth codes is defined as
$$\left(\begin{array}{cccc} c_{0}\left(x\right) & c_{1}\left(x\right) & \cdots & c_{n-1}\left(x\right) \end{array}\right)\cdot H_{r\times n}^{T}\equiv 0\;\left(\mathrm{mod}\;M_{p}\left(x\right)\right),$$
where $H_{r\times n}$ is the r×n parity-check matrix
$$H_{r\times n}=\left[\begin{array}{ccccc} 1 & 1 & 1 & \cdots & 1\\ 1 & x & x^{2} & \cdots & x^{n-1}\\ \vdots & \vdots & \vdots & \ddots & \vdots \\ 1 & x^{\left(r-1\right)} & x^{(r-1)2} & \cdots & x^{\left(r-1)(n-1\right)} \end{array}\right],$$
and 0 is the all-zero row of length *r*. We denote the Blaum–Roth codes defined above as C(p,n,r). When p≥n and *p* is a prime number, we can always retrieve all the information bits from any *k* out of the *n* polynomials [17], i.e., C(p,n,r) are MDS codes.

If we let $c_{p-1,j}=0$ for all j=0,1,…,n−1, then C(p,n,r) can be equivalently defined as the following p·r linear constraints. (The subscripts are taken as modulo *p* unless otherwise specified.)
$$\sum_{j=0}^{n-1}c_{{\langle m-\ell \cdot j\rangle }_{p},j}=0,$$
where 0≤m≤p−1 and 0≤ℓ≤r−1.

Suppose that the λ columns ${\left\{e_{i}\right\}}_{i=0}^{\lambda -1}$ are erased, where λ≥2 and $0\leq e_{0}<\cdots <e_{\lambda -1}<n$. Let the δ=n−λ surviving columns be ${\left\{h_{j}\right\}}_{j=0}^{\delta -1}$, where $0\leq h_{0}<\cdots <h_{\delta -1}<n$ and ${\left\{e_{i}\right\}}_{i=0}^{\lambda -1}\cup {\left\{h_{j}\right\}}_{j=0}^{\delta -1}=\left\{0,1,\ldots ,n-1\right\}$. We have
(1)$$\left(\begin{array}{cccc} c_{e_{0}}\left(x\right) & c_{e_{1}}\left(x\right) & \cdots & c_{e_{\lambda -1}}\left(x\right) \end{array}\right)\cdot V_{\lambda \times \lambda }^{T}=S,$$
over the ring $R_{p}$, where $V_{\lambda \times \lambda }$ is the λ×λ square
$$V_{\lambda \times \lambda }=\left[\begin{array}{ccccc} 1 & 1 & 1 & \cdots & 1\\ x^{e_{0}} & x^{e_{1}} & x^{e_{2}} & \cdots & x^{e_{\lambda -1}}\\ \vdots & \vdots & \vdots & \ddots & \vdots \\ x^{\left(\lambda -1\right)e_{0}} & x^{\left(\lambda -1\right)e_{1}} & x^{\left(\lambda -1\right)e_{2}} & \cdots & x^{\left(\lambda -1\right)e_{\lambda -1}} \end{array}\right],$$
and $S=\left(\begin{array}{cccc} S_{0}\left(x\right) & S_{1}\left(x\right) & \cdots & S_{\lambda -1}\left(x\right) \end{array}\right)$, where the λ syndrome polynomials are
(2)$$S_{\ell }\left(x\right)=\sum_{j=0}^{\delta -1}x^{\ell \cdot h_{j}}c_{h_{j}}\left(x\right)\;\mathrm{for}\;0\leq \ell \leq \lambda -1.$$

In this paper, we present three efficient decoding methods to solve the linear systems in Equation (Equation 1) over the ring $F_{2}\left[x\right]/\left(1+x^{p}\right)$.

### 2.2. Efficient Operations over $F_{2}\left[x\right]/\left(1+x^{p}\right)$

It is more efficient to compute the multiplication of a power of *x* and division of the factor $1+x^{b}$ over the ring $F_{2}\left[x\right]/\left(1+x^{p}\right)$ rather than over the ring $R_{p}$: (i) Let $a\left(x\right)\in R_{p}$, and the multiplication $x^{i}\cdot a\left(x\right)$ over the ring $R_{p}$ in [17] (Equation (19)) takes p−1 XORs, while the multiplication $x^{i}\cdot a\left(x\right)$ over the ring $F_{2}\left[x\right]/\left(1+x^{p}\right)$ takes no XORs [20]. (ii) Let $g\left(x\right),f\left(x\right)\in F_{2}\left[x\right]/\left(1+x^{p}\right)$, where *d* is a positive integer, which is coprime with *p*. Consider the equation
(3)$$\left(1+x^{d}\right)g\left(x\right)\equiv f\left(x\right)\;\left(\mathrm{mod}\;1+x^{p}\right),$$
where f(x) has an even number of nonzero terms. Given such f(x) and *d*, we can compute g(x) by Lemma 1.

**Lemma** **1.**
*[Lemma 8] in [21] The coefficients of g(x) in Equation (Equation 3) are given by*

$$\begin{array}{cc} g_{p-1} & =0,\;g_{p-d-1}=f_{p-1},\;g_{d-1}=f_{d-1},\\ g_{p-(\ell +1)d-1} & =g_{p-\ell d-1}+f_{p-\ell d-1}\;\mathrm{for}\;\ell =1,2,\ldots ,p-3. \end{array}$$



By Lemma 1, computing the division $\frac{f(x)}{1+x^{d}}$ takes p−3 XORs, but we are not sure whether g(x) has an even number of nonzero terms or not. If we want to guarantee that g(x) has an even number of nonzero terms, we should use Lemma 2 to compute the division $\frac{f(x)}{1+x^{d}}$.

**Lemma** **2.**
*[Lemma 13] in [20] The coefficients of g(x) in Equation (Equation 3) are given by*

$$\begin{array}{cc} g_{0} & =f_{2d}+f_{4d}+\cdots +f_{(p-1)d},\\ g_{\ell d} & =g_{(\ell -1)d}+f_{\ell d}\;\mathrm{for}\;\ell =1,2,\ldots ,p-1. \end{array}$$



By Lemma 2, the division $\frac{f(x)}{1+x^{d}}$ takes $\frac{3p-5}{2}$ XORs, and g(x) has an even number of nonzero terms. However, computing the division $\frac{f(x)}{1+x^{d}}$ in [Corollary 2] in [17] takes 2(p−1) XORs over the ring $R_{p}$, which is strictly larger than the decoding methods in Lemmas 1 and 2. It is shown in [Theorem 5] in [19] that we can always solve the equations in Equation (Equation 1) over the ring $F_{2}\left[x\right]/\left(1+x^{p}\right)$ of which all the solutions are congruent to each other after modulo $M_{p}\left(x\right)$. Therefore, we can first solve the equations in Equation (Equation 1) over the ring $F_{2}\left[x\right]/\left(1+x^{p}\right)$ and then obtain the unique solution by taking modulo $M_{p}\left(x\right)$ to reduce the computational complexity.

### 2.3. Multiple Multiplications over $F_{2}\left[x\right]/\left(1+x^{p}\right)$

Note that in our modified syndrome-based decoding method and the modified interpolation-based decoding method, we need to compute multiple polynomial multiplications, where each polynomial has two nonzero terms. Suppose that we want to compute the following *m* multiplications
(4)$$L\left(x^{\tau }\right)=\prod_{i=0}^{m-1}\left(x^{\tau }-x^{\xi_{i}}\right)\;\left(\mathrm{mod}\;1+x^{p}\right),$$
where *m* is a positive integer, 0≤τ≤p−1 such that $\tau \notin \{\xi_{0},\xi_{1},\ldots ,\xi_{m-1}\}$, and $0\leq \xi_{0}<\cdots <\xi_{m-1}<n$.

We can derive from Equation (Equation 4) that
(5)$$L\left(x^{\tau }\right)=x^{\pi }\cdot \prod_{i=0}^{m-1}\left(1+x^{d_{i}}\right)\;\left(\mathrm{mod}\;1+x^{p}\right),$$
where $\pi =\sum_{i=0}^{m-1}\min\left(\tau ,\xi_{i}\right)$ modulo *p* and $d_{i}=\left|\tau -\xi_{i}\right|$ for i=0,1,…,m−1.

Algorithm 1 presents a method to simplify the multiplications in Equation (Equation 4). In Algorithm 1, we use $\Gamma_{\ell }$ to denote the number of the polynomial $1+x^{\ell }$ in the multiplication $L(x^{\tau })$. Note that we only need to count the number of $1+x^{\ell }$ for $1\leq \ell \leq \frac{p-1}{2}$, because the equation $1+x^{\ell }\equiv x^{\ell }\cdot \left(1+x^{p-\ell }\right)$ modulo $1+x^{p}$ holds for $\frac{p-1}{2}<\ell <n$. If $\Gamma_{\ell }>1$, then we have ${\left(1+x^{\ell }\right)}^{\Gamma_{\ell }}={\left(1+x^{\ell }\right)}^{\Gamma_{\ell }-2\left\lfloor \frac{\Gamma_{\ell }}{2}\right\rfloor }\cdot {\left(1+x^{2\ell }\right)}^{\left\lfloor \frac{\Gamma_{\ell }}{2}\right\rfloor }$. Therefore, we can always merge $\Gamma_{\ell }$ multiplications ${\left(1+x^{\ell }\right)}^{\Gamma_{\ell }}$ into $\Gamma_{\ell }-\left\lfloor \frac{\Gamma_{\ell }}{2}\right\rfloor$ multiplications and the computational complexity can be reduced with Algorithm 1. When Algorithm 1 is executed, all elements of count-array Γ should be zero or one, and the length η of the final $L(x^{\tau })$ is between 1 and *m*.
**Algorithm 1:** Simplify the multiple multiplications.
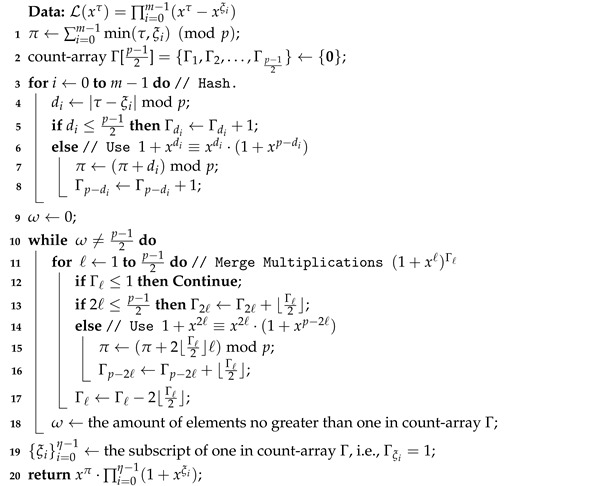


## 3. Decoding Algorithm

In this section, we present two decoding methods over the ring $F_{2}\left[x\right]/\left(1+x^{p}\right)$ by modifying two existing decoding methods [17,18] that can reduce the decoding complexity.

Recall that the λ erased columns are λ columns ${\left\{e_{i}\right\}}_{i=0}^{\lambda -1}$, and the δ=n−λ surviving columns are δ columns ${\left\{h_{j}\right\}}_{j=0}^{\delta -1}$.

### 3.1. Modified Syndrome-Based Method

We define the function of the indeterminate *z*
$$G_{i}\left(z\right)=\prod_{s=0,\neq i}^{\lambda -1}\left(1-x^{e_{s}}z\right)=\sum_{\ell =0}^{\lambda -1}G_{i,\ell }\left(x\right)z^{\ell },$$
and the syndrome function $S\left(z\right)=\sum_{\ell =0}^{r-1}S_{\ell }\left(x\right)z^{\ell }$, where 0≤i≤λ−1 and $S_{\ell }\left(x\right)$ is given in Equation (Equation 2). We can obtain in [Equation (18)] in [17] that
$$\begin{array}{cc} \prod_{s=0,\neq i}^{\lambda -1}\left(x^{e_{i}}-x^{e_{s}}\right)c_{e_{i}}\left(x\right)\; & \equiv \sum_{\ell =0}^{\lambda -1}G_{i,\lambda -1-\ell }\left(x\right)S_{\ell }\left(x\right)\\ \; & \equiv \sigma_{i}\left(x\right)\;\left(\mathrm{mod}\;M_{p}\left(x\right)\right). \end{array}$$

Therefore, the $\sigma_{i}\left(x\right)$ can be regarded as the coefficient of $z^{\lambda -1}$ of the polynomial $G_{i}\left(z\right)S\left(z\right)$. Then, the erased column $c_{e_{i}}\left(x\right)$ is given by $\frac{\sigma_{i}\left(x\right)}{\prod_{s=0,\neq i}^{\lambda -1}\left(x^{e_{i}}-x^{e_{s}}\right)}$, where 0≤i≤λ−1.

Note that the terms of set ${\left\{S_{\ell }\left(x\right)z^{\ell }\right\}}_{\ell =\lambda }^{r-1}$ are not involved in computing the coefficient of $z^{\lambda -1}$ of the polynomial $G_{i}\left(z\right)S\left(z\right)$. Thus, we can just consider the first λ terms (the λ coefficients of degrees less than λ) of S(z) when computing these coefficients, but all the *r* terms of S(z) are calculated in [Step 1] in [17]. This is one essential way our modified syndrome-based decoding method obtains a lower decoding complexity than the original method in [17].

Moreover, the syndrome polynomials $S_{\ell }\left(x\right)$ satisfy
(6)$$S_{0}\left(1\right)=S_{1}\left(1\right)=\cdots =S_{\lambda -1}\left(1\right),$$
i.e., the λ syndrome polynomials $S_{\ell }\left(x\right)$ either all have an even number of nonzero terms, or they all have an odd number of nonzero terms, from the definition of Equation (Equation 2).

Let $G\left(z\right)=\left(1-x^{e_{i}}z\right)G_{i}\left(z\right)$ and Q(z)=G(z)S(z). Then, we have
(7)$$\begin{array}{cc} Q(z) & =\left(1-x^{e_{i}}z\right)\prod_{s=0,\neq i}^{\lambda -1}\left(1-x^{e_{s}}z\right)S\left(z\right)\\ \; & =\prod_{s=0}^{\lambda -1}\left(1-x^{e_{s}}z\right)S\left(z\right)=\sum_{\ell =0}^{r+\lambda -1}Q_{\ell }\left(x\right)z^{\ell }. \end{array}$$

Thus, Q(z) is independent of the erasure index *i*, and we only need to compute Q(z) once in the decoding procedure. Recall that $\sigma_{i}\left(x\right)$ is the coefficient of $z^{\lambda -1}$ of the polynomial $G_{i}\left(z\right)S\left(z\right)$; then, the $\sigma_{i}\left(x\right)$ is also the coefficient of $z^{\lambda -1}$ of the polynomial $\frac{Q(z)}{\left(1-x^{e_{i}}z\right)}=\frac{\left(1-x^{e_{i}}z\right)G_{i}\left(z\right)S\left(z\right)}{\left(1-x^{e_{i}}z\right)}$ for all 0≤i≤λ−1. Suppose that
$$\frac{Q(z)}{\left(1-x^{e_{i}}z\right)}=f_{0}^{i}\left(x\right)+f_{1}^{i}\left(x\right)z+\cdots +f_{\lambda -1}^{i}\left(x\right)z^{\lambda -1}+\cdots ,$$
we can derive the recurrence formula
(8)$$f_{\ell }^{i}\left(x\right)=\left\{\begin{array}{cc} Q_{0}\left(x\right), & \ell =0;\\ x^{e_{i}}\cdot f_{\ell -1}^{i}\left(x\right)+Q_{\ell }\left(x\right), & \ell >0; \end{array}\right.$$
where 0≤i≤λ−1. Notice that $\sigma_{i}\left(x\right)=f_{\lambda -1}^{i}\left(x\right)$ holds. Similar to S(z), we only compute the first λ terms (the λ coefficients of degrees less than λ) of Q(z), since the other coefficients of Q(z) are not needed, but all the r+λ terms of Q(z) are calculated in [Step 2] in [17]. This is another way our modified syndrome-based decoding method obtains a lower decoding complexity than the original method in [17]. Algorithm 1 shows our modified syndrome-based decoding method over the ring $F_{2}\left[x\right]/\left(1+x^{p}\right)$.

The following Lemma shows that we can always compute the divisions in steps 11–12 of Algorithm 2 by Lemmas 1 and 2 when λ≥2.

**Lemma** **3.**
*In steps 11–12 of Algorithm 2, the $\sigma_{i}\left(x\right)$ has an even number of nonzero terms for all 0≤i≤λ−1, and we can employ Lemmas 1 and 2 to compute the divisions.*


**Proof.** From Equation (Equation 8) and steps 7–10 of Algorithm 2, we obtain
$$\sigma_{i}\left(x\right)=x^{\left(\lambda -1\right)e_{i}}Q_{0}\left(x\right)+x^{\left(\lambda -2\right)e_{i}}Q_{1}\left(x\right)+\cdots +Q_{\lambda -1}\left(x\right),$$
where 0≤i≤λ−1. If the number of polynomials in the set ${\left\{Q_{j}\left(x\right)\right\}}_{j=0}^{\lambda -1}$, which has an odd number of nonzero terms, is an even number, then the $\sigma_{i}\left(x\right)$ has an even number of nonzero terms for 0≤i≤λ−1. In the following, we will show this is true. According to Equation (Equation 6) and step 3 of Algorithm 2, $Q_{0}\left(1\right)=\cdots =Q_{\lambda -1}\left(1\right)$ holds.

**Firstly**, we consider $Q_{0}\left(1\right)=\cdots =Q_{\lambda -1}\left(1\right)=1$. We denote the λ polynomials ${\left\{Q_{j}\left(x\right)\right\}}_{j=0}^{\lambda -1}$ with ε=0,1,…,λ as ${\left\{Q_{j}^{\varepsilon }\left(x\right)\right\}}_{j=0}^{\lambda -1}$. Let $Q_{j}^{0}\left(x\right)$ be the initial $Q_{j}\left(x\right)$ for 0≤j≤λ−1.

To prove that the number of polynomials with an odd number of nonzero terms in the set ${\left\{Q_{j}^{\varepsilon }\left(x\right)\right\}}_{j=0}^{\lambda -1}$ is even, it is equivalent to prove that $\sum_{j=0}^{\lambda -1}Q_{j}^{\varepsilon }\left(1\right)=0$.

**Algorithm 2:** Modified syndrome-based decoding method.

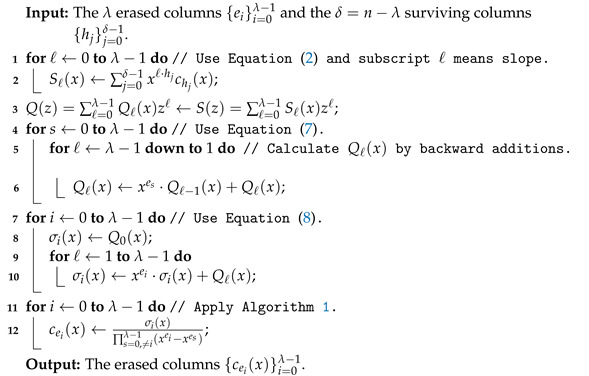



According to Equation (Equation 7) and steps 4–6 of Algorithm 2, we have
(9)$$Q_{j}^{\varepsilon }\left(1\right)=\left\{\begin{array}{cc} Q_{j}^{\varepsilon -1}\left(1\right), & j=0;\\ Q_{j-1}^{\varepsilon -1}\left(1\right)+Q_{j}^{\varepsilon -1}\left(1\right), & 1\leq j\leq \lambda -1; \end{array}\right.$$
where ε=1,2,…,λ. The $Q_{j}^{1}\left(1\right)=0$ holds for all j≥1. We can obtain by induction
(10)$$Q_{j}^{\varepsilon }\left(1\right)=Q_{j-1}^{\varepsilon -1}\left(1\right)+Q_{j}^{\varepsilon -1}\left(1\right)=0\;\mathrm{for}\;\mathrm{all}\;j\geq \varepsilon \geq 1.$$

Note that $\sum_{j=0}^{\lambda -1}Q_{j}^{2}\left(1\right)=0$; we can suppose that there are an even number of polynomials in the set ${\left\{Q_{j}^{\varepsilon }\left(x\right)\right\}}_{j=0}^{\lambda -1}$, which has an odd number of nonzero terms, when ε=y≥2, i.e., $\sum_{j=0}^{\lambda -1}Q_{j}^{y}\left(1\right)=0$ first. We have $\sum_{j=0}^{\lambda -1}Q_{j}^{y+1}\left(1\right)$; so,
(11)$$\begin{array}{cc} \sum_{j=0}^{\lambda -1}Q_{j}^{y+1}\left(1\right)\; & =Q_{0}^{y}\left(1\right)+\sum_{j=1}^{\lambda -1}(Q_{j-1}^{y}\left(1\right)+Q_{j}^{y}\left(1\right))\\ \; & =\sum_{j=0}^{\lambda -1}Q_{j}^{y}\left(1\right)+\sum_{j=0}^{\lambda -2}Q_{j}^{y}\left(1\right)\\ \; & =Q_{\lambda -1}^{y}\left(1\right)=0. \end{array}$$

Equation (Equation 11) comes from Equation (Equation 10) with j=λ−1. Therefore, there are an even number of polynomials in the set ${\left\{Q_{j}^{y+1}\left(x\right)\right\}}_{j=0}^{\lambda -1}$, which has an odd number of nonzero terms.

**Secondly**, when $Q_{0}\left(1\right)=\cdots =Q_{\lambda -1}\left(1\right)=0$, the argument is similar. This completes the proof.  □

According to Lemma 3, we can use Lemmas 1 and 2 to compute the divisions in step 12. The number of divisions required in step 12 is recorded as $L_{i}$, which ranges from 1 to λ−1 for i=0,1,…,λ−1. So, we can obtain $c_{e_{i}}\left(x\right)$ in step 12 by recursively computing the division $L_{i}$ times, while the number of nonzero terms of the polynomial resulting from the first $L_{i}-1$ divisions is even. Therefore, we can execute these divisions by Lemma 2 and execute the last division by Lemma 1. The computational complexity $T_{D}$ in steps 11–12 of Algorithm 2 is
(12)$$T_{D}=\sum_{i=0}^{\lambda -1}(\left(L_{i}-1\right)\frac{3p-5}{2}+p-3),$$
where $\lambda \left(p-3\right)\leq T_{D}\leq \lambda \left(\lambda -2\right)\frac{3p-5}{2}+\lambda \left(p-3\right)$.

In steps 11–12 of Algorithm 2, we take the λ(λ−1) division without Algorithm 1, in which λ divisions are executed by Lemma 1 and λ(λ−2) divisions are executed by Lemma 2; however, the number of the divisions can be reduced with Algorithm 1. In Table 1, we show the average number of divisions in steps 11–12 of Algorithm 2 executed by Lemma 1 and Lemma 2 with Algorithm 1 for (p,n)∈{(5,5),(7,7)}.

We specify the computational complexity of Algorithm 2 as follows:Steps 1–2 take λ(δ−1)p=λ(n−λ−1)p XORs.Steps 3–6 take λ(λ−1)p XORs.Steps 7–10 take λ(λ−1)p XORs.Steps 11–12 take $T_{D}$ XORs by Equation (Equation 12).

Then, the computational complexity $T_{Alg\;2}$ of Algorithm 2 is
(13)$$T_{Alg\;2}=\lambda \left(n+\lambda -3\right)p+T_{D},$$
where
$$p\lambda^{2}+(\left(n-2\right)p-3)\lambda \leq T_{Alg\;2}\leq \frac{5(p-1)}{2}\lambda^{2}+(\left(n-5\right)p+2)\lambda .$$

Recall that the computational complexity of the decoding method in [17] is
$$\frac{7p-4}{2}\lambda^{2}-\frac{7p-2}{2}\lambda +r\left(n-1\right)p.$$
which is strictly larger than $T_{Alg\;2}$.

Table 2 evaluates the computational complexity of the decoding method in [17] and Algorithm 2 for some parameters. The results in Table 2 demonstrate that Algorithm 2 has much lower decoding complexity, compared with the original decoding method in [17]. For example, Algorithm 2 has 40.60% less decoding complexity than the decoding method in [17] when (p,n,r)=(7,7,4),λ=3.

The reason why Algorithm 2 has lower decoding complexity than the decoding method in [17] can be summarized as the following three points.

**Firstly**, we only consider the first λ terms (the λ coefficients of degrees less than λ) for both S(z) and Q(z) in computing the coefficients of $z^{\lambda -1}$, while all *r* terms of S(z) and all r+λ terms of Q(z) are calculated in the decoding method in [17], where r≥λ.

**Secondly**, all the divisions in Algorithm 2 are executed over the ring $F_{2}\left[x\right]/\left(1+x^{p}\right)$ by Lemmas 1 and 2, which takes p−3 XORs and $\frac{3p-5}{2}$ XORs for each division, respectively. In addition, the division in [17] is executed over the ring $R_{p}$, which takes 2(p−1) XORs [17] (Corollary 2).

**Thirdly**, we apply Algorithm 1 to steps 11–12 of Algorithm 2, which can significantly reduce the number of divisions, thus reducing the number of XORs required.

### 3.2. Modified Interpolation-Based Decoding Method

According to the decoding method in [18], we can recover the erased column $c_{e_{i}}\left(x\right)$ with 0≤i≤λ−1 by
(14)$$c_{e_{i}}\left(x\right)=\sum_{j=0}^{\delta -1}c_{h_{j}}\left(x\right)\frac{f_{i}\left(x^{h_{j}}\right)}{f_{i}\left(x^{e_{i}}\right)}\;\left(\mathrm{mod}\;M_{p}\left(x\right)\right),$$
where $f_{i}\left(y\right)=\prod_{s=0,\neq i}^{\lambda -1}\left(y-x^{e_{s}}\right)$ and $f\left(y\right)=\prod_{s=0}^{\lambda -1}\left(y-x^{e_{s}}\right)$. Let
(15)$$a_{j}\left(x\right)=c_{h_{j}}\left(x\right)\cdot f\left(x^{h_{j}}\right)=\prod_{s=0}^{\lambda -1}\left(x^{h_{j}}-x^{e_{s}}\right)\cdot c_{h_{j}}\left(x\right)\;\left(\mathrm{mod}\;M_{p}\left(x\right)\right),$$
where 0≤j≤δ−1. Then, $a_{j}\left(x\right)$ has an even number of nonzero terms, and we only need to compute once for $a_{j}\left(x\right)$ in the decoding procedure, since $a_{j}\left(x\right)$ is independent of the erasure index *i*. Let
(16)$$\begin{array}{cc} b_{i}\left(x\right) & =\sum_{j=0}^{\delta -1}\frac{a_{j}\left(x\right)}{x^{h_{j}}-x^{e_{i}}}\;\left(\mathrm{mod}\;M_{p}\left(x\right)\right), \end{array}$$
(17)$$\begin{array}{cc} c_{e_{i}}\left(x\right) & =\frac{b_{i}\left(x\right)}{f_{i}\left(x^{e_{i}}\right)}=\frac{b_{i}\left(x\right)}{\prod_{s=0,\neq i}^{\lambda -1}\left(x^{e_{i}}-x^{e_{s}}\right)}\;\left(\mathrm{mod}\;M_{p}\left(x\right)\right), \end{array}$$
where 0≤i≤λ−1, and $M_{p}\left(x\right)=1+x+\cdots +x^{p-1}$. Algorithm 3 shows our modified interpolation-based method over the ring $F_{2}\left[x\right]/\left(1+x^{p}\right)$.

After using Algorithm 1, the number of polynomial multiplications in step 2 ranges from 1 to λ. Thus, the computational complexity $T_{M}$ in steps 1–2 of Algorithm 3 is
(18)$$\left(n-\lambda \right)p\leq T_{M}\leq \left(n-\lambda \right)\lambda p.$$

**Algorithm 3:** Modified interpolation-based method.

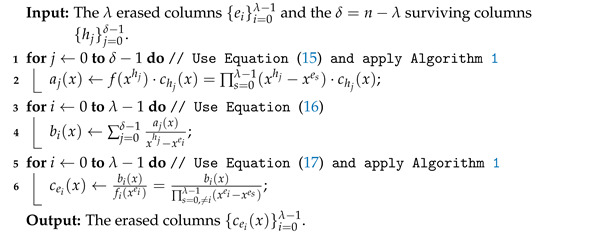



In steps 1–2, we need to take λ multiplications without Algorithm 1, which takes (n−λ)λp XORs; however, with Algorithm 1, the number of multiplications involved in steps 1–2 can be reduced. In Table 3, we show the average number of XORs involved in steps 1–2 of Algorithm 3 with Algorithm 1 for (p,n)∈{(5,5),(7,7)}. The results in Table 3 show that we can reduce the number of XORs with Algorithm 1, especially for a large value of λ.

Only steps 4 and 6 of Algorithm 3 are needed to compute the division. We should employ Lemma 2 to execute the divisions in steps 3–4 in Algorithm 3, since $b_{i}\left(x\right)$ in step 6 of Algorithm 3 should have an even number of nonzero terms. Notice that steps 5–6 of Algorithm 3 are exactly the same as steps 11–12 of Algorithm 2.

We specify the computational complexity of Algorithm 3 as follows:Steps 1–2 require $T_{M}$ XORs by Equation (Equation 18).Steps 3–4 need λ(δ−1) additions and λδ divisions by Lemma 2, which require $\lambda \left(n-\lambda -1\right)p+\lambda \left(n-\lambda \right)\frac{3p-5}{2}$ XORs in total.Steps 5–6 require $T_{D}$ XORs by Equation (Equation 12).

Then, the computational complexity of Algorithm 3 is
(19)$$T_{Alg\;3}=T_{M}+\lambda \left(n-\lambda -1\right)p+\lambda \left(n-\lambda \right)\frac{3p-5}{2}+T_{D},$$
where
$$-\frac{5(p-1)}{2}\lambda^{2}+\left(\frac{5n-2}{2}p-\frac{5}{2}n-3\right)\lambda +np\leq T_{Alg\;3}\leq -2p\lambda^{2}+\left(\frac{7n-6}{2}p-\frac{5}{2}n+2\right)\lambda .$$

Recall that the computational complexity of the decoding method in [18] is
$$\left(-2p+1\right)\lambda^{2}+(4\left(n-1\right)p-3n+4)\lambda +n\left(p-1\right),$$
which is larger than that of our Algorithm 3.

Table 4 evaluates the computational complexity of the decoding method in [18] and Algorithm 3 for some parameters. The results in Table 4 demonstrate that our Algorithm 3 had much lower decoding complexity, compared with the original decoding method in [18]. For example, Algorithm 3 had a 34.13% lower decoding complexity than the decoding method in [18], when (p,n,r)=(7,7,4),λ=3.

The reason why Algorithm 3 has a lower decoding complexity than that of the decoding method in [18] is summarized as follows.

**Firstly**, all the divisions in Algorithm 3 were executed over the ring $F_{2}\left[x\right]/\left(1+x^{p}\right)$ by Lemmas 1 and 2, which used p−3 XORs and (3p−5)/2 XORs for each division, respectively. The division in the decoding method in [18] was executed over the ring $R_{p}$, which used 2(p−1) XORs.

**Secondly**, we applied our Algorithm 1 to steps 1–2 and steps 5–6, which significantly reduced the number of multiplications, thus reducing the number of XORs required.

## 4. LU Decomposition-Based Method

The LU factorization of a matrix [22] is to express the matrix as a product of a lower triangular matrix L and an upper triangular matrix U. According to the LU factorization of the Vandermonde matrix [23], we can express a Vandermonde matrix as a product of several lower triangular matrices and several upper triangular matrices. Therefore, we can solve the Vandermonde linear equations by first solving the linear equations with the encoding matrices that are the upper triangular matrices and then solving the linear equations with the encoding matrices that are the lower triangular matrices.

Suppose that the λ erased columns are λ columns ${\left\{e_{i}\right\}}_{i=0}^{\lambda -1}$ and the δ=n−λ surviving columns are ${\left\{h_{j}\right\}}_{j=0}^{\delta -1}$. Algorithm 4 shows our LU decomposition-based method over the ring $F_{2}\left[x\right]/\left(1+x^{p}\right)$.

According to [Theorem 8] in [19], Equation (Equation 1) can be factorized into
(20)$$\left(\begin{array}{cccc} c_{e_{0}}\left(x\right) & c_{e_{1}}\left(x\right) & \cdots & c_{e_{\lambda -1}}\left(x\right) \end{array}\right)\cdot \left(L_{\lambda }^{\left(1\right)}L_{\lambda }^{\left(2\right)}\cdots L_{\lambda }^{\left(\lambda -1\right)}\right)\cdot \left(U_{\lambda }^{\left(\lambda -1\right)}U_{\lambda }^{\left(\lambda -2\right)}\cdots U_{\lambda }^{\left(1\right)}\right)=S,$$
over the ring $R_{p}$, where $U_{\lambda }^{\left(\theta \right)}$ is the upper triangle matrix
(21)$$U_{\lambda }^{\left(\theta \right)}=\left[\begin{array}{cc} I_{\lambda -\theta -1} & 0\\ ine0 & \begin{array}{cccccc} 1 & x^{e_{0}} & 0 & \cdots & 0 & 0\\ 0 & 1 & x^{e_{1}} & \cdots & 0 & 0\\ \vdots & \vdots & \vdots & \ddots & \vdots & \vdots \\ 0 & 0 & 0 & \cdots & 1 & x^{e_{\theta -1}}\\ 0 & 0 & 0 & \cdots & 0 & 1 \end{array} \end{array}\right],$$
and $L_{\lambda }^{\left(\theta \right)}$ is the lower triangle matrix
(22)$$L_{\lambda }^{\left(\theta \right)}=\left[\begin{array}{cc} I_{\lambda -\theta -1} & 0\\ ine0 & \begin{array}{ccccc} 1 & 0 & \cdots & 0 & 0\\ 1 & x^{e_{\lambda -\theta }}+x^{e_{\lambda -\theta -1}} & \cdots & 0 & 0\\ \vdots & \vdots & \ddots & \vdots & \vdots \\ 0 & 0 & \cdots & x^{e_{\lambda -2}}+x^{e_{\lambda -\theta -1}} & 0\\ 0 & 0 & \cdots & 1 & x^{e_{\lambda -1}}+x^{e_{\lambda -\theta -1}} \end{array} \end{array}\right],$$
for θ=1,2,…,λ−1.   
**Algorithm 4:** LU decomposition-based method.
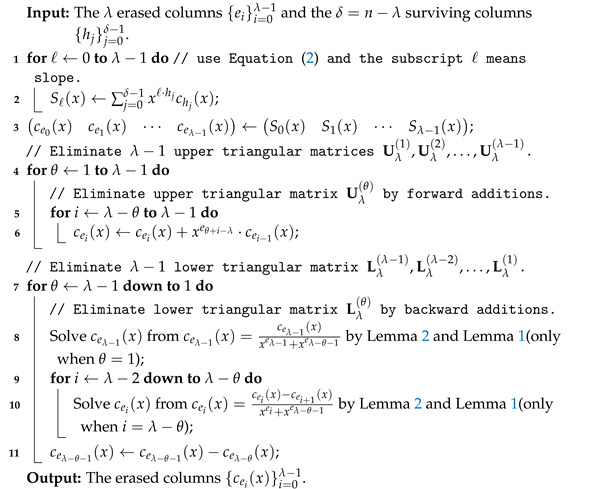


We specify the computational complexity of Algorithm 4 as follows:Steps 1–2 require λ(δ−1)p=λ(n−λ−1)p XORs.Steps 3–11 require λ(λ−1)p+(λ−1)(p−3)+(λ−1)(λ−2)(3p−5)/4 XORs at most, according to [Theorem 10] in [19].

Then, the computational complexity of Algorithm 4 is
(23)$$T_{Alg\;4}=\frac{3p-5}{4}\lambda^{2}+\frac{(4n-13)p+3}{4}\lambda +\frac{p+1}{2}.$$

## 5. Comparison and Conclusions

Table 5 evaluates the decoding complexity of Algorithm 2–4 for some parameters. The results of Table 5 demonstrate that Algorithm 2 performs better than Algorithm 3 if $\lambda \leq \frac{n}{2}$; otherwise, if $\lambda >\frac{n}{2}$, then Algorithm 3 has less decoding complexity. Algorithm 4 has less decoding complexity than both Algorithms 2 and 3, when λ is small. However, when λ is large, Algorithm 3 is more efficient than Algorithm 4. For example, compared with Algorithm 2–4 have 21.98% and 40.66% less decoding complexity, respectively, when (p,n,r)=(5,5,4),λ=3.

In this paper, we presented three efficient decoding methods for the erasures of Blaum–Roth codes that all have lower decoding complexity than the existing decoding methods. The efficient implementation of the proposed decoding methods in practical storage systems is one of our future works.

## Figures and Tables

**Table 1 entropy-24-01499-t001:** The average number of XORs involved in steps 11–12 of Algorithm 2.

p, n	*λ*	Without Algorithm 1			Apply Algorithm 1			Improvement(%)
		Lemma 2	Lemma 1	XORs	Lemma 2	Lemma 1	XORs	
(5, 5)	2	0	2	4	0	2	4	0%
	3	3	3	21	2	3	16	23.81%
	4	8	4	48	0	4	8	83.33%
(7, 7)	2	0	2	8	0	2	8	0%
	3	3	3	36	2.4	3	31.2	13.33%
	4	8	4	80	4.4	4	51.2	36%
	5	15	5	140	1	5	28	80%
	6	24	6	216	6	6	72	66.67%

**Table 2 entropy-24-01499-t002:** Decoding complexity of method in [17] and Algorithm 2.

p, n, r	*λ*	XORs in [17]	XORs of $T_{\mathrm{Alg}2}$	Improvement(%)
(5, 5, 3)	2	89	44	50.56%
	3	150	91	39.33%
(7, 7, 4)	2	211	92	56.40%
	3	300	178.2	40.60%
	4	434	275.2	36.59%

**Table 3 entropy-24-01499-t003:** The average number of XORs involved in steps 1–2 of Algorithm 3.

p, n	*λ*	Without Algorithm 1		Apply Algorithm 1		Improvement(%)
		Multiplication	XORs	Multiplication	XORs	
(5, 5)	2	6	30	5	25	16.67%
	3	6	30	2	10	66.67%
	4	4	20	2	10	50%
(7, 7)	2	10	70	9	63	10%
	3	12	84	8.4	58.8	30%
	4	12	84	3.6	25.2	70%
	5	10	70	4	28	60%
	6	6	42	3	21	50%

**Table 4 entropy-24-01499-t004:** Decoding complexities of the decoding method in [18] and our Algorithm 3.

p, n, r	*λ*	XORs in [18]	XORs of $T_{\mathrm{Alg}\;3}$	Improvement(%)
(5, 5, 3)	2	122	79	35.25%
	3	146	71	51.37%
(7, 7, 4)	2	292	207	29.11%
	3	378	249	34.13%
	4	438	228.4	47.85%

**Table 5 entropy-24-01499-t005:** Decoding complexities of the proposed three decoding methods.

p, n, r	*λ*	Total XORs			$\frac{T_{\mathrm{Alg}\;2}-T_{\mathrm{Alg}\;3}}{T_{\mathrm{Alg}\;2}}$	$\frac{T_{\mathrm{Alg}\;2}-T_{\mathrm{Alg}\;4}}{T_{\mathrm{Alg}\;2}}$
		$T_{\mathrm{Alg}\;2}$	$T_{\mathrm{Alg}\;3}$	$T_{\mathrm{Alg}\;4}$		
(5, 5, 4)	2	44	79	32	−79.55%	27.27%
	3	91	71	54	21.98%	40.66%
	4	128	38	81	70.31%	36.72%
(7, 7, 6)	2	92	207	74	−125%	19.57%
	3	178.2	249	121	−39.73%	32.10%
	4	275.2	228.4	176	17.01%	36.05%
	5	343	171	239	50.15%	30.32%
	6	492	141	310	71.34%	36.99%

## Data Availability

All data generated or analysed during this study are included in this published article.

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
