# Peer review of "Three Efficient All-Erasure Decoding Methods for Blaum–Roth Codes"

_entropy, 2022, doi:10.3390/e24101499_

Round 1

Reviewer 1 Report

This paper presents three decoding methods for Blaum-Roth Codes. These decoding methods have lower decoding complexity than existing decoding methods: syndrome-based decoding method and interpolation-based decoding method. The key ideas to simplify complexity are replacing Mp(x) with 1+x^p during division operation and reducing some operations during the decoding procedure. I feel the contribution is sufficient and can be published. Here are my recommendations for improvement:

1.  It will be easier to be understood if give more details about the related concepts and background of LU decomposition-based method before the pseudocode in section 4.

2. It would be better to mention more state-of-art works, e.g. graftage coding and so on.

3.  Pay attention to the spelling of some words.

“we present an alogorithm…with lower complexity” on page 2;

“two non-zero term” should be corrected to “two non-zero terms”;

“code-word” is usually written as “codeword”;

“The results in Table 2…original decoding method in [1]” on page 8

“Algorithm 2 have much lower deocoding complexity” should be corrected to “Algorithm 2 has much lower decoding complexity

Reviewer 2 Report

This paper proposes three decoding methods which have lower complexity than known results for Blaum-Roth codes, named Modified Syndrome-based Method, Modified Interpolation-based Decoding Method, and LU Decomposition-based Method. The key of reducing complexity is the methods in this paper are taken modulo 1+x^p rather than M_p(x)=1+x+...+x^{p-1}. 

The methods are interesting and organized, and the result is improved. But I found some typos in my reading, see below for instance.

1.      15, The storage system just replace damaged disks: replace -> replaces

2.      42, The essential reason of why our modified decoding: delete “of”

3.     56, which have two non-zero term:  term->terms

4.      125, The number of polynomials in the set which have odd number of nonzero terms is even is equivalent to that the (Equation):  Grammar is wrong.

5.     132, in which lambda divisions is executed by Lemma 1 and lambda*(lambda-2) divisions is executed by: is ->are

6.      203, Algorithm 3 is more efficient Algorithm 4: efficient than Algorithm 4.

       7. 209, one of our future work: work->works
